# The Spatial Niche and Influencing Factors of Desert Rodents

**DOI:** 10.3390/ani14050734

**Published:** 2024-02-27

**Authors:** Xin Li, Na Zhu, Ming Ming, Lin-Lin Li, Fan Bu, Xiao-Dong Wu, Shuai Yuan, He-Ping Fu

**Affiliations:** 1College of Grassland, Resources and Environment, Inner Mongolia Agricultural University, 29 Erdos East Street, Saihan District, Hohhot 010011, China; lixin9623@163.com (X.L.); jvn0397@163.com (N.Z.); bai_mingming2023@163.com (M.M.); lilinlin9630@emails.imau.edu.cn (L.-L.L.); bufan@emails.imau.edu.cn (F.B.); wuxiaodong_hgb@163.com (X.-D.W.); 2Key Laboratory of Grassland Rodent Ecology and Rodent Pest Control, Universities of Inner Mongolia Autonomous, Hohhot 010011, China; 3Key Laboratory of Grassland Resources, Ministry of Education, 29 Erdos East Street, Hohhot 010011, China

**Keywords:** *Orientallactaga sibirica*, *Meriones meridianus*, *Dipus sagitta*, *Phodopus roborovskii*, spatial niche, impact factors, coexistence

## Abstract

**Simple Summary:**

Rodents in the Alxa desert area (Inner Mongolia, China) face more human disturbance (including grazing, reclamation, excavation of wild medicinal materials, etc.) than in other areas. We used the capture-mark-recapture method to continuously monitor the four main rodent spatial niches from 2017 to 2021. At the same time, we monitored the vegetation, soil, temperature, and humidity in the survey area. The results showed that the spatial niche breadth of rodents was mainly affected by population density (rodents) and shrubs (height and density) in the habitat. The coexistence strategy of the northern three-toed jerboa (*Dipus sagitta*) with midday gerbil (*Meriones meridianus*) and desert hamster (*Phodopus roborovskii*) is based on the difference in body size and foraging strategy. The five-toed jerboa (*Orientallactaga sibirica*) promotes coexistence through spatial niche separation. Rodents can make a trade-off between foraging efficiency and the cost of travel to achieve coexistence.

**Abstract:**

Resource partitioning may allow species coexistence. Sand dunes in the typical steppe of Alxa Desert Inner Mongolia, China, consisting of desert, shrub, and grass habitats, provide an appropriate system for studies of spatial niche partitioning among small mammals. In this study, the spatial niche characteristics of four rodents, *Orientallactaga sibirica*, *Meriones meridianus*, *Dipus sagitta,* and *Phodopus roborovskii*, and their responses to environmental changes in the Alxa Desert were studied from 2017 to 2021. Using the capture-mark-recapture method, we tested if desert rodents with different biological characteristics and life history strategies under heterogeneous environmental conditions allocate resources in spatial niches to achieve sympatric coexistence. We investigated the influence of environmental factors on the spatial niche breadth of rodents using random forest and redundancy analyses. We observed that the spatial niche overlap between *O. sibirica* and other rodents is extremely low (overlap index ≤ 0.14). *P. roborovskii* had the smallest spatial niche breadth. Spatial niche overlap was observed in two distinct species pairs, *M. meridianus* and *D. sagitta*, and *P. roborovskii* and *D. sagitta*. The Pielou evenness index of rodent communities is closely related to the spatial distribution of rodents, and the concealment of habitats is a key factor affecting the spatial occupation of rodents.

## 1. Introduction

The concept of an ecological niche is fundamental to the study of coexistence [1,2,3]. Broadly, a niche refers to the ideal set of conditions and settings for an animal to maximize fitness. Animals can adjust their niche breadth by adaptation or behavioral changes in inter-specific competition to reduce the intensity of competition between species [4]. Spatial niche is one of the important dimensions of niche. It is the basis for understanding the coexistence and interaction of mammalian species [5]. If the spatial niche overlap of two species is too high, it indicates that their required environmental resources (food, shelter, and reproduction) are highly similar [6], which will inevitably aggravate the competition between the two species [7,8,9]. When shared resources are limited, individuals compete [10]. If spatial niches of two species are not separated, the interaction between species may reduce the fitness of a species and eventually lead to the local extinction of species [11,12]. Habitat separation can be used as a mechanism for species coexistence, especially when food resources are limited [13]. Differences in spatial use can effectively promote coexistence [14]. Therefore, quantifying the overlap and separation of species’ spatial ecological niches may aid in understanding the resource allocation patterns of sympatric coexisting species [15]. At the regional level, the quality and distribution of patch resources play a significant role in the spatial distribution of dominant animals [16,17].

Rodents are highly sensitive to their environment, serving as important indicators of environmental changes and key species in the food web. Studies of rodents have been key to the development of desert ecology [18]. The spatial niche of rodents is determined by the species’ adaptability to habitats and their utilization of environmental resources. The heterogeneity of habitats allows for the coexistence of a greater variety of species [19]. The spatial niche of species is significantly influenced by both biotic and abiotic factors, such as climate change [20], food availability [21], inter-specific competition [1,22,23], predation risk [1,24], vegetation [25], and landscape complexity [26]. Thus, the spatial niche of rodents is not static but rather changes with the seasons and the characteristics of the vegetation in their habitat. The study of the spatial ecological niche of rodents should also consider factors such as population dynamics, behavior, spatial distribution, habitat productivity, and habitat safety as related factors to comprehensively assess the dynamic changes in their spatial ecological niche [27].

Our study system included the midday gerbil (*Meriones meridianus*, 49.53–53.6 g), desert hamster (*Phodopus roborovskii*, 20.4–27.5 g), five-toed jerboa (*Orientallactaga sibirica*, 89.2–96.2 g), and northern three-toed jerboa (*Dipus sagitta*, 59.6–89.7 g), which are common species in many East Asian deserts [28,29]. These species are nocturnal. *M. meridianus* and *P. roborovskii* are quadrupedal, central-place foragers with food-hoarding habits [30]; *P. roborovskii* has cheek pouches, which can increase harvest rates without raising predation costs [31]. In contrast, *D. sagitta* and *O. sibirica* are bipedal wanderers with a much larger home range and generally do not store food [32]. Like many other jerboas, *D. sagitta* and *O. sibirica* are also known for their bipedal locomotory gait, which may increase their ability to travel long distances and chances of escaping predators in open Microhabitats. *D. sagitta* and *O. sibirica* hibernate in winter (usually from late October to early April), while *M. meridianus* and *P. roborovskii* rely on the food collected in autumn to survive the tough times [30,33].

The biological characteristics and life history of the four rodents are quite different. How do they adapt to the harsh desert environment and achieve stable coexistence? A mechanism of coexistence on a local scale normally requires an axis of heterogeneity and a trade-off, such that each species can perform better than its competitors along some part of the axis [34]. In light of this, this study analyzes species’ spatial niches, coexistence patterns, and their influencing factors in the desert rodent community to explore the impact of habitat fragmentation on the coexistence and diversity maintenance mechanisms of desert rodents. We hypothesize that the habitat conditions and abundance of food resources are key factors affecting the rodents’ habitat selection. Coexistence in the same microhabitat is possible if rodents can make a trade-off between foraging efficiency and the cost of travel.

## 2. Materials and Methods

### 2.1. Study Area

This study was conducted in the southern Alxa Desert, Inner Mongolia, China (E 104°10′–105°30′, N 37°24′–38°25′). Our study area has a continental climate with cold and dry winters and warm summers. Annual precipitation ranges from 75 mm to 215 mm, about 70% of which falls from June to September. The soil is grey desert soil and grey-brown soil. The vegetation is sparse, and the plants are mainly xerophytic, super xerophytic, and halophytic shrubs. *M. meridianus, D. sagitta*, *P. roborovskii,* and *O. sibirica* are dominant small-mammal species [35]. Other natural enemies include *Bubo bubo*, *Vormela peregusna*, and *Vulpes corsac* [36]. In total, 24 sampling plots were selected in typical areas of human disturbance (grazing and land reclamation) in the southern desert area of Alxa Left Banner. Field surveys were conducted in spring (April), summer (July), and autumn (October) from 2017 to 2021 (Figure 1).

### 2.2. Data Collection

#### 2.2.1. Rodent Data Collection

To capture rodent individuals, a 7 × 8 m trapping grid (1 ha) at a 15 m inter-trap distance was established at the center of each plot (60 ha) (Figure 2). We placed one wire-mesh live trap (42 cm × 17 cm × 13 cm, Guixi Rodent Equipment Co., Ltd., Guixi, China) at each grid intersection (henceforth trap station). Traps were baited with fresh peanuts and checked twice (morning and afternoon) each day. We recorded the species name and the capture location of the captured individuals. Each captured individual was injected under the pelage with a passive integrated transponder (PIT) (2.12 mm × 8 mm, Guangzhou Ruimai Intelligent Technology Co., Ltd., Guangzhou, China) tag with a unique identification number (ID). In order to prevent rainfall and other sudden events from disturbing trapped individuals, a wooden box (15 cm × 10 cm × 10 cm) was placed in the live trap to protect the rodents entering the cage [36]. We live-trapped rodents for four consecutive days in each season. A total of 80,640 cage·day were placed. We calculated the population quantity of rodents captured according to the minimum number known alive method [37].

#### 2.2.2. Environmental Factor Data Collection

Vegetation sampling: We randomly selected three 100 m^2^ plots within each unit to sample shrubs in each season. Within each 100 m^2^ plot, we randomly placed three 1 m^2^ quadrats to sample grasses and forbs. We measured the height, cover, density, abundance, and biomass of shrubs and herbs [36].

Soil sampling: At the same time as plant sampling, soil data were collected, including soil moisture content and soil hardness at 0–5 cm, 5–10 cm, 10–15 cm, and 15–20 cm. The soil hardness was measured using a soil hardness meter (TYD-2, Zhejiang Top Yunnan Technology Co., Ltd., Zhejiang, China), and the soil moisture content was determined using a drying method (105 °C, 8 h).

Meteorological sampling: The meteorological data come from the “Luanjingtan Meteorological Station in Alxa League”, which is 5.95 ± 0.34 (Mean ± SE) km away from the study area. The meteorological data include the monthly average temperature, monthly average humidity, and monthly sunshine duration indicators for the whole year from 2017 to 2021.

### 2.3. Statistical Analyses

We used the minimum number known alive (MNKA) method to calculate the population density of rodents [37,38].
MNKA = *a* + *b*,(1)
where *a* is the actual number caught in the cage deployment; *b* is the number of previously marked individuals caught after cage deployment but not at this cage deployment. Rodent population density data were tested using the Shapiro–Wilk method and were found to be abnormally distributed (*p* < 0.05). The data were processed by log (*n* + 1). We used the one-way ANOVA to compare the means of four rodent population densities.

We used the Shannon–Wiener niche breadth index to estimate spatial niche breadth [39].
*Bi* = 1/*lgr* × [*lg*∑*N_ij_* − (1/∑*N_ij_*) (∑*N_ij_*/*lgN_ij_*)](2)
where *B_i_* is the niche breadth of species *i*, *r* is the resource level (the survey plot is defined as the resource level), *N_ij_* is the value of the resource level *j* used by species *i* (*j* represents the mark-recapture plot, *i* represents the average number of captured in the sample area), and *lg* is the logarithm to base 10. The value of *B_i_* ranges between 0 and 1.

The spatial niche breadth index of the main species of desert rodents in different seasons was analyzed using the two-way repeated measures ANOVA using SPSS Statistics 26.0. All data were tested using the Shapiro–Wilk method and were found to be normally distributed (*p* > 0.05). According to the sphericity test (Machly W = 0.631, *p* = 0.031 < 0.05), it does not conform to the sphericity test. The correction results in a one-way analysis of variance ‘Greenhouse-Geisser‘ were taken as the standard.

We calculated the spatial niche overlap index for each pair of rodent species using Colwell and Futuyma’s niche overlap index (1971) [40].
*O_ik_* = 1 − ½ × ∑|*N_ij_*/*N_i_* − *N_kj_*/*N_k_*|(3)
where *O_ik_* is the niche overlap index between species *i* and species *k*; *N_ij_* is the value of species *i* appeared in the *j*th resource levels; *N_i_* is the value of species *i* appeared in all resource levels; *N_kj_* is the value of species k appeared in the *j*th resource levels; *N_k_* is the value of species *k* appeared in all resource levels. The value of *O_ik_* ranges between 0 and 1 [40].

We used 5 indicators to estimate habitat productivity and ‘concealment’, including average height (*AH*), density of shrub (*Den*), average coverage of shrub (*C*), coverage of shrub (*TC.S*), and plant biomass (*TB*).
*AH* = (*LH* + *MH* + *SH*)/3(4)
*Den* = *IN*/*100* m^2^(5)
*C* = (*3.14* × *SR^2^* × *Den*)/*100* m^2^(6)
*TC.S* = ∑*Ci*(7)
*TB* = (*P* × *DW* × *Den*)/100(8)
where *AH* represents the average height (cm) of shrubs, and *LH*, *MH*, and *SH* represent the height (cm) of large, medium, and small shrubs, respectively. *Den* represents density. *IN* represents the number of individuals. *C* represents the average coverage (%) of a shrub on a per unit area basis. *SR* represents the radius of the shrub canopy (cm). *TC* represents the total coverage (%) of shrubs in a unit area. *S* represents the number of species. *Ci* represents the coverage (%) of the *i* th shrub on a unit area. *TB* represents the biomass (g/m^2^) of shrubs in a unit area. *P* represents the sampling proportion of the current year’s branches of individual plants, and *DW* represents the dry weight of biomass for each species, with a drying time greater than 30 days. The measurement accuracy is 0.01 g. The determination of grasses indicators includes the average height (cm), density (plants/m^2^), and dry weight (accuracy 0.01 g) of grasses in a 1 m^2^ quadrat [38,41].

There are 31 environmental factors included in this analysis, including 7 biotic factors and 24 abiotic factors (Table 1). We used a random forest model [42] to screen out the key environmental factors affecting the change in niche breadth. According to the results, the importance of variables is sorted by histogram. ariables greater than the mean value “Mean” are considered to have high contribution or importance, and variables greater than the maximum value “MAX” are considered to have very high contribution or importance. We extracted the factors in front of “Max”. Collinearity between environmental factors is evaluated by calculating the variance inflation factor (VIF), with a threshold of 10; when VIF ≥ 10, there is strong multicollinearity. After removing factors with strong multicollinearity, the data in the species factor variable group were transformed using the Hellinger transformation. The data of environmental factor variables were standardized, and the relationship between environmental factor variables and the spatial niche of the four rodent species was analyzed using the vegan package for redundancy analysis (RDA) in R 4.2.0. Using the ordiR2step function in the vegan package [43], we performed forward selection to identify the best environmental factors that explain the spatial niche variation of rodents.

## 3. Results

### 3.1. Population Density of Rodents in Desert Areas

The population density of *D. sagitta*, *O. sibirica,* and *M. meridianus* were significantly higher in spring and summer than in autumn (*D. sagitta*: *F* _(1,358)_ = 24.56, *p* < 0.001; *O. sibirica*: *F*
_(1,358)_ = 18.56, *p* < 0.001; *M. meridianus*: *F*
_(1,358)_ = 10.71, *p* < 0.01). There was no seasonal difference in the population density of *P. roborovskii* (*F*
_(1,358)_ = 1.024, *p* > 0.05) (Figure 3).

### 3.2. Spatial Niche Characteristics of Rodents in Desert Areas

#### 3.2.1. Spatial Niche Breadth of Rodents in Desert Areas

The spatial niche breadth of *D. sagitta*, *O. sibirica,* and *M. meridianus* in the desert region did not significantly vary between seasons (*D. sagitta*: *F*
_(2,12)_ = 1.473, *p* > 0.05; *O. sibirica*: *F*
_(2,12)_ = 1.199, *p* > 0.05; *M. meridianus*: *F*
_(2,12)_ = 1.024, *p* > 0.05). In autumn, the spatial niche breadth of *P. roborovskii* was significantly lower than that in spring (*F*
_(2,12)_ = 6.060, *p* < 0.05). The spatial niche breadth of *P. roborovskii* was significantly smaller than that of the other three rodents in all seasons (Spring: *F*
_(3,16)_ = 10.288, *p* < 0.01; Summer: *F*
_(3,16)_ = 13.536, *p* < 0.001; Autumn: *F*
_(3,16)_ = 11.622, *p* < 0.001) (Table 2).

#### 3.2.2. Spatial Niche overlap of Rodents in Desert Areas

From the perspective of spatial niche overlap, the niche overlap index between the species pair *M. meridianus* and *D. sagitta* and the species pair *P. roborovskii* and *D. sagitta* was high in three of four seasons, ranging from 0.26 to 0.55, indicating significant competition among the three species in the spatial dimension. The niche overlap index between *O. sibirica* and the other three species was low in all three seasons, ranging from 0.04 to 0.14, indicating that *O. sibirica* used spatial niche separation to reduce competition with the other three species (Table 3).

### 3.3. Screening of Key Factors Affecting the Spatial Niche of Rodents

We observed that key environmental factors affecting the spatial niche of *D. sagitta* were population densities of *O. sibirica* and *D. sagitta*, monthly average temperature, and population size of the *M. meridianus,* Pielou index of grasses, soil moisture, and soil hardness in the 5–10 cm layer (Figure 4A). The key environmental factors affecting the spatial niche of *O. sibirica* are the population density of *O. sibirica*, the height of shrubs, and the population size of *M. meridianus* (Figure 4B). The key environmental factors that affect the spatial niche breadth of *P. roborovskii* are the population size of *P. roborovskii*, Shannon–Wiener index of grasses, Shannon–Wiener index of shrubs, shrub coverage, and shrub density (Figure 4C). The key environmental factors that affect the spatial niche breadth of *M. meridianus* are its own population size, shrub height, Pielou index of rodents, *D. sagitta* population size, shrub coverage, shrub biomass, Simpson index of shrub, *O. sibirica* population size, and monthly sunshine hours (Figure 4D).

### 3.4. Response of Spatial Niche of Rodents to Environmental Changes in Desert Area

A total of 19 biotic and abiotic factors were screened out based on the random forest model to potentially affect the spatial niche variation of rodents and those factors with a very high contribution. After removing environmental factors with strong collinearity, a total of 17 environmental variables were included in the environmental factor variable group. Redundancy analysis (RDA) was used to analyze the relationship between 17 environmental factors and the spatial niche of four rodent species. The detrended correspondence analysis (DCA) indicates that the length of the first axis of the spatial niche breadth is less than 3 (0.65), so this study applies the redundancy analysis based on linear models. The results of the RDA indicate that the first two canonical axes can explain 91.84% of the total variance, of which the first ordering axis alone explains 56.94%. The proportion of total variance explained by all canonical axes after adjustment is *R*^2^_adj_ = 0.5274 (Figure 5). The permutation test results show that the model has an ideal sorting effect (*F* = 2.9037, *p* = 0.001). The results of the forward selection model for environmental factors suggest that the height of shrubs in the habitat, the Simpson index of shrubs, the population size of *P. roborovskii*, the population size of *M. meridianus*, and the Pielou index of rodents are the main factors affecting the spatial niche breadth of rodents (Table 4). The *P. roborovskii* prefers to choose low-growing and dense shrubs. Shrub diversity has a negative effect on two jerboas (*O. sibirica* and *D. sagitta*). The increase in the population size of *M. meridianus* will increase the spatial niche breadth of *O. sibirica* and *P. roborovskii* and decrease the niche breadth of *M. meridianus* and *D. sagitta*. The population size of *P. roborovskii* was significantly negatively correlated with the spatial niche breadth of *M. meridianus, D. sagitta,* and *O. sibirica* (Figure 5).

## 4. Discussion

The spatial niche breadth of *M. meridianus* was always the widest among the three seasons, followed by *D. sagitta* and *O. sibirica*, while the spatial niche breadth of *P. roborovskii* was the narrowest. The spatial niche breadth of the four rodents was the widest in summer. Niche breadth is a comprehensive reflection of the resources that a species can utilize. Species with a wider niche tend to be less specialized and rely on certain resources [44,45]. The *M. meridianus* in our study area has a wide distribution range and strong adaptability and tends to be a generalized species in spatial distribution [46]. Given that each of these four species is nocturnal, we assume that foraging time is lower in the summer than in other seasons with longer nights. Accordingly, we have chosen to interpret the larger (but insignificant) spatial niche breadth in summer to mean that rodents are driven to forage broader regions in search of food, given their shorter nighttime to forage [47].

The degree of spatial niche overlap between *O. sibirica* and the other three species of rodents showed a gradual decrease with seasonal changes. Spatial niche overlap reflects the geographical differences in species distribution and the degree of interspecific competition [48]. The decrease in the population of a certain species in a community is one of the key factors in reducing competition and achieving sympatric coexistence [49]. From spring to autumn, the population size of *O. sibirica* gradually decreases, and in winter, it enters hibernation. Hibernation is the result of rodents’ comprehensive weighing of environmental temperature and resource availability [3,50,51,52]. The decrease in population size and hibernation have alleviated competition between *O. sibirica* and other rodent species, which is beneficial for coexistence.

The spatial niche overlap between *P. roborovskii* and *D. sagitta*, *M. meridianus,* and *D. sagitta* is relatively high. The reason why the spatial niche overlap between the two pairs of species, *P. roborovskii* and *D. sagitta*, *M. meridianus,* and *D. sagitta*, is high but stable is not only due to the hibernating behavior of *D. sagitta* but also due to differences in body size and foraging strategies. Smaller species tend to have higher foraging efficiency, while larger species can always find resource-rich patches faster than smaller species [53,54]. The Alxa Desert region has been proven to be a typical assemblage community [55]. The habitat resources are patchily distributed, and *D. sagitta* is adept at moving between multiple foraging patches to select higher-quality patches, but their foraging efficiency is low [56]. Conversely, both *M. meridianus* and *P. roborovskii* are species with high foraging efficiency but poor long-distance transmission capabilities. Therefore, the coexistence strategy of *D. sagitta* with *M. meridianus* and *P. roborovskii* is based on differences in body size and foraging strategies. This result is consistent with Brown et al.’s coexistence theory [49].

The shelter conditions of the habitat influence the spatial distribution of rodents. *O. sibirica* and *D. sagitta* prefer habitats with low-density and tall shrubs. Many documents have suggested that bipedal jerboas prefer open habitats [56]. The dense vegetation condition is not conducive to the jumping of jerboa, which increases the difficulty of feeding and avoiding predators [30,36,39,56]. Yuan Shuai and his colleagues analyzed rodents and environmental factors in desert areas using structural equation modeling. They found that plant cover had a negative effect on the bipedal activity of *O. sibirica* but a positive effect on the quadrupedal activity of *M. meridianus* [38]; this is consistent with our results. *P. roborovskii* prefers habitats with a high diversity of shrubs, which is related to its diet. *P. roborovskii* mainly feeds on plant seeds, accounting for about 2/3 of its food composition [57]. The more species of shrubs, the more food resources they can provide.

The studies by Weihui Dong and Yu Ji have shown that the population dynamics of *O. sibirica* in the Inner Mongolia desert region are mainly influenced by other coexisting rodent species, especially the dominant rodent species, and are negatively correlated with the population density of the dominant rodent species [58,59]. This was consistent with our observations. The variation in the spatial niche breadth of *O. sibirica* is mainly influenced by the population size of *M. meridianus*. The spatial distribution of *P. roborovskii* was influenced by both its own population size and the population density of coexisting species [59]. Competition with *M. meridianus* drives the *P. roborovskii* to increase its spatial niche breadth. The niche variation hypothesis proposes that as food resources become limited, competition within and between species increases, and the variability in the degree of dietary specialization among individuals increases, increasing the niche breadth of the population [60]. The changes in the spatial niche breadth of *O. sibirica* and *P. roborovskii* support the niche variation hypothesis [2,61], as the spatial niche breadth of the two species increases with inter and intra-specific competition, making fuller use of environmental resources and facilitating their coexistence in desert areas.

This study has potential limitations. We used the minimum number known alive (MNKA) method to calculate the population quantity of rodents. This method is commonly used to assess population size with capture-mark-recapture data. However, MNKA uses information from prior and subsequent capture sessions to assess the population at each point in a longitudinal study. Therefore, it is subject to negative bias that is greatest at the beginning and end of the study and least in the middle. Stochastic simulations performed with constant population size and capture rate showed that MNKA peaked in the middle of the study. The tapering bias was greatest when the survival rate between capture sessions was high [62]. We also conducted a capture-mark-recapture experiment in 2022. When MNKA was used to calculate the number of rodent populations in 2021, the 2022 capture data was also included in the statistics. However, there are still some errors in rodent population estimation. We hope that this method can be improved in future studies.

## 5. Conclusions

Spatial niche separation is one of the reasons for the coexistence of rodents in desert areas. Their coexistence strategy, based on their functional attributes and life history characteristics, involves balancing between shrub and open habitats, foraging efficiency and foraging costs, foraging efficiency and maintenance costs to reduce competition and achieve coexistence.

## Figures and Tables

**Figure 1 animals-14-00734-f001:**
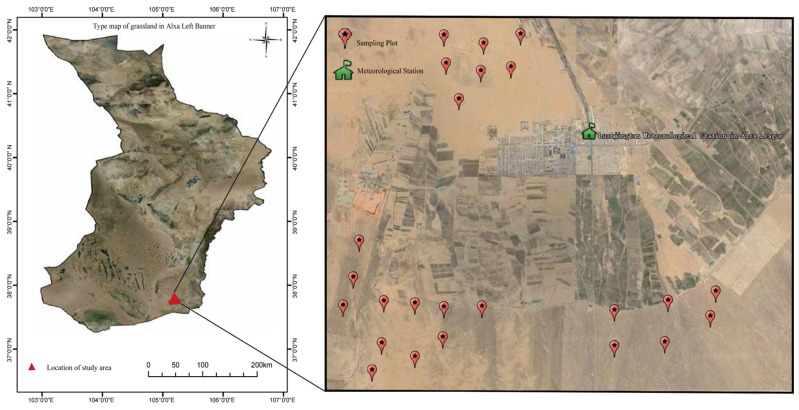
Geographical location of the study area and the position of sampling plots.

**Figure 2 animals-14-00734-f002:**
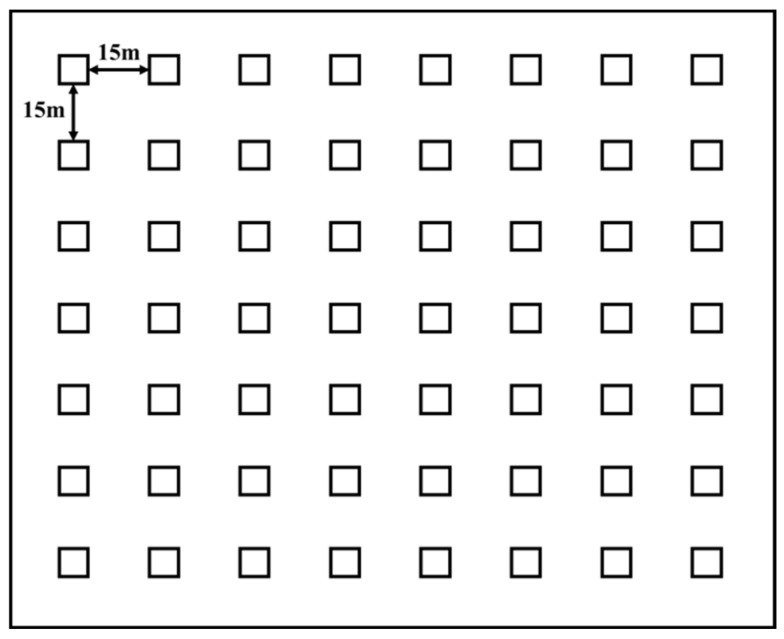
Location map of cages.

**Figure 3 animals-14-00734-f003:**
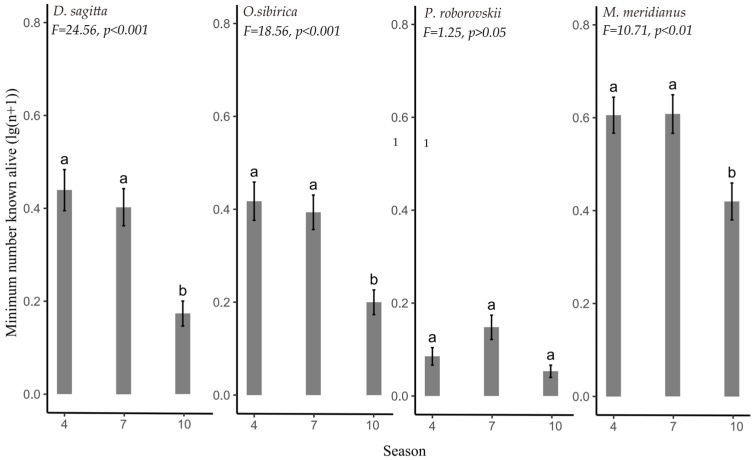
Population density of desert rodents in different seasons. Different lowercase letters indicate (a and b) significant inter-seasonal differences in the spatial niche of each rodent.

**Figure 4 animals-14-00734-f004:**
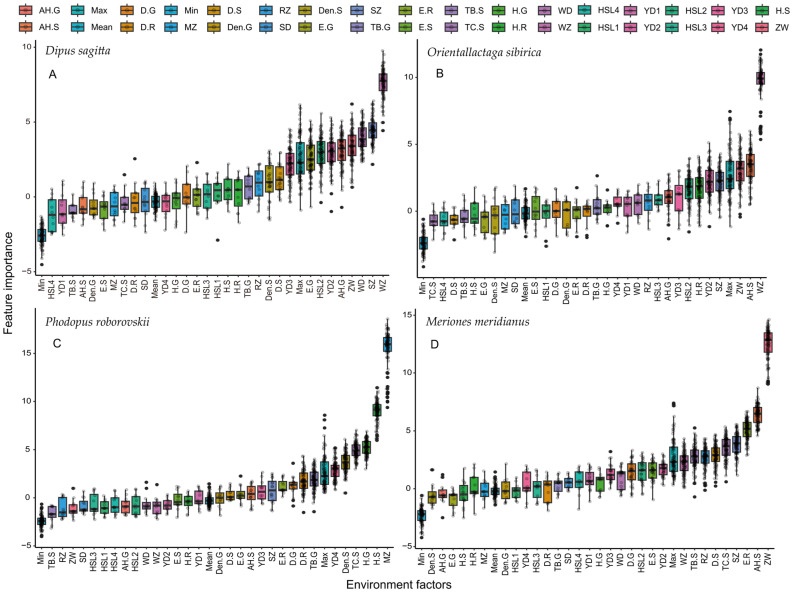
Key environmental factors affecting the spatial niche width of rodents. (**A**) Key environmental factors affecting the spatial niche width of *D. sagitta*, (**B**) key environmental factors affecting the spatial niche width of *O. sibirica*, (**C**) key environmental factors affecting the spatial niche width of *P. roborovskii*, (**D**) key environmental factors affecting the spatial niche width of *M. meridianus*. Variables greater than the mean value “Mean” are considered to have high contribution or importance, and variables greater than the maximum value “MAX” are considered to have very high contribution or importance.

**Figure 5 animals-14-00734-f005:**
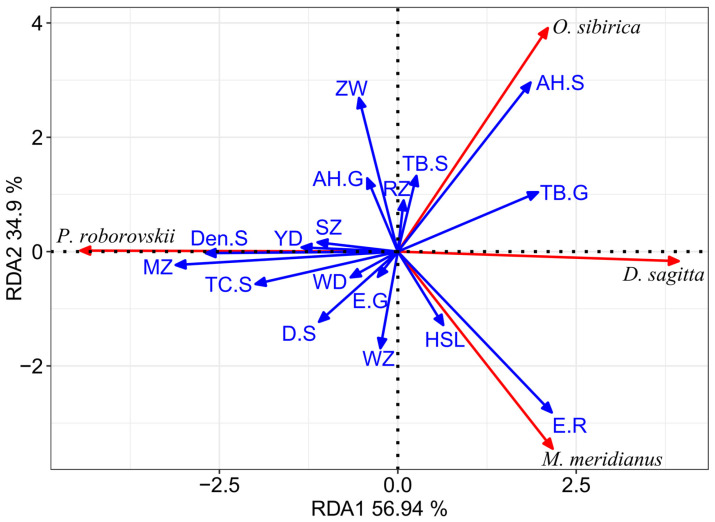
RDA ordination diagram of the rodent spatial niche breadth and environmental factors in desert areas.

**Table 1 animals-14-00734-t001:** All environmental factors included in the analysis.

Environmental Factor	Acronym	Environmental Factor	Acronym
Simpson diversity index of rodents	D.R	Shannon-Wiener index of shrub	H.S
Pielou diversity index of rodents	E.R	Soil water content of 0–5 cm	HSL1
Shannon–Wiener index of rodents	H.S	Soil water content of 5–10 cm	HSL2
Population of *M. meridianus*	ZW	Soil water content of 10–15 cm	HSL3
Population of *D. sagitta*	SZ	Soil water content of 15–20 cm	HSL4
Population of *P. roborovskii*	MZ	Soil hardness of 0–5 cm	YD1
Population of *O. sibirica*	WZ	Soil hardness of 5–10 cm	YD2
Average height of grasses	AH.G	Soil hardness of 10–15 cm	YD3
Average height of a shrub	AH.S	Soil hardness of 15–20 cm	YD4
Simpson diversity index of grasses	D.G	Biomass of grasses	TB.G
Simpson diversity index of shrub	D.S	Biomass of shrub	TB.S
Density of grasses	Den.G	Coverage of shrub	TC.S
Density of shrub	Den.S	Monthly mean relative humidity	SD
Pielou diversity index of grasses	E.G	Monthly mean relative temperature	WD
Pielou diversity index of shrub	E.S	Monthly mean sunshine duration	RZ
Shannon–Wiener index of grasses	H.G		

**Table 2 animals-14-00734-t002:** Spatial niche breadth index (mean ± se) of main species desert rodents in different seasons. Different capital letters (A and B) indicate significant interspecies differences in the spatial niche of each rodent, and different lowercase letters indicate (a and b) significant inter-seasonal differences in the spatial niche of each rodent.

Method	Species	Spring (Mean ± se)	Summer (Mean ± se)	Autumn (Mean ± se)	Season
Simple Effects	*D. sagitta*	0.70 ± 0.06 Aa	0.74 ± 0.03 Aa	0.65 ± 0.01 Aa	*F*_(2,12)_ = 1.473, *p* > 0.05, *η*^2^ = 0.160
	*O. sibirica*	0.71 ± 0.05 Aa	0.75 ± 0.02 Aa	0.67 ± 0.0 Aa	*F*_(2,12)_ = 1.199, *p* = 0.309, *η*^2^ = 0.145
	*P. roborovskii*	0.42 ± 0.07 Bab	0.51 ± 0.06 Ba	0.32 ± 0.09 Bb	*F*_(2,12)_ = 6.060, *p* < 0.05, *η*^2^ = 0.447
	*M. meridianus*	0.85 ± 0.03 Aa	0.85 ± 0.05 Aa	0.76 ± 0.06 Aa	*F*_(2,12)_ = 1.024, *p* = 0.383, *η*^2^ = 0.120
	Species	*F*_(3,16)_ = 10.288, *p* < 0.01, *η*^2^ = 0.659	*F*_(3,16)_ = 13.536, *p* < 0.001, *η*^2^ = 0.717	*F*_(3,16)_ = 11.622, *p* < 0.001, *η*^2^ = 0.685	—
Repeated Measures	Species	*F* = 29.866, *p* < 0.001, *η*^2^ = 0.848			
	Season	*F* = 5.141, *p* < 0.05, *η*^2^ = 0.243			
	Species × Season	*F* = 0.314, *p* > 0.05, *η*^2^ = 0.056			

**Table 3 animals-14-00734-t003:** Spatial niche overlap index (mean ± se) of main species of desert rodents in different seasons.

Season	Species	*D. sagitta*	*O. sibirica*	*P. roborovskii*	*M. meridianus*
Spring	*D. sagitta*	1			
	*O. sibirica*	0.18 ± 0.02	1		
	*P. roborovskii*	0.33 ± 0.10	0.14 ± 0.07	1	
	*M. meridianus*	0.45 ± 0.05	0.14 ± 0.07	0.17 ± 0.07	1
Summer	*D. sagitta*	1			
	*O. sibirica*	0.16 ± 0.01	1		
	*P. roborovskii*	0.55 ± 0.04	0.07 ± 0.03	1	
	*M. meridianus*	0.45 ± 0.03	0.07 ± 0.03	0.28 ± 0.06	1
Autumn	*D. sagitta*	1			
	*O. sibirica*	0.11 ± 0.05	1		
	*P. roborovskii*	0.35 ± 0.09	0.04 ± 0.04	1	
	*M. meridianus*	0.26 ± 0.043	0.04 ± 0.04	0.16 ± 0.05	1

**Table 4 animals-14-00734-t004:** RDA analysis and forward selection of the impact of environmental factors on rodent spatial niche.

Environment Factor	Full Models				Forward Selection		
	RDA1	RDA2	*r^2^*	*p*	*r^2^*	*F*	*p*
WD	−0.836	−0.549	0.020	0.739			
RZ	0.119	0.993	0.024	0.715			
AH.G	−0.315	0.949	0.054	0.483			
TB.G	0.889	0.458	0.157	0.101			
E.G	−0.563	−0.826	0.009	0.889			
AH.S	0.562	0.827	0.377	0.003 **	0.173	10.323	0.005
Den.S	−1.000	−0.032	0.228	0.03 *			
TB.S	0.225	0.974	0.055	0.476			
TC.S	−0.961	−0.278	0.136	0.117			
D.S	−0.691	−0.723	0.085	0.298	0.059	4.535	0.030
HSL	0.452	−0.892	0.061	0.457			
YD	−1.000	0.029	0.057	0.450			
SZ	−0.994	0.114	0.040	0.591			
WZ	−0.173	−0.985	0.088	0.294			
MZ	−0.996	−0.093	0.308	0.007 **	0.268	10.266	0.001
ZW	−0.188	0.982	0.222	0.037 *	0.067	4.571	0.023
E.R	0.626	−0.780	0.374	0.003 **	0.123	5.471	0.012

* indicates significant level, *p* < 0.05; ** indicates *p* < 0.01.

## Data Availability

The data used in this study are available from the first author upon reasonable request.

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
