# Peer review of "The Spatial Niche and Influencing Factors of Desert Rodents"

_animals, 2024, doi:10.3390/ani14050734_

Round 1

Reviewer 1 Report

Comments and Suggestions for Authors

The abundance of four rodent species was measured over five years in a desert. To determine the species coexistence, the niche breath of each species was calculated. The niche was characterized by vegetation structure, soil characteristics and interspecific competition. Habitat selection and behavioral ecology of the four rodent species were discussed according to the niche overlap and the eco-morphology of each species.

Although the information may be sufficient to support the conclusions, the process of combining the field data to produce the estimates of niche breadth is not sufficiently explained. For example, the experimental design included 24 replicates of sampling points that were visited several times in the spring, summer and autumn of five years. Thus, there is spatial replication and repeated annual and seasonal measurements. However, the methods and results do not explain in sufficient detail how spatial variability and temporal repetition were combined in the calculation of the results. The results of population size estimates for each rodent species are also not included. These are aspects that can be improved and require further work.

There is an excess of triviality in the Introduction that needs to be corrected. For example, some sentences are trivial, e.g., "The spatial niche of species is significantly influenced by both biotic and abiotic factors..." (line 67), "quality and distribution of patch resources play a significant role in the spatial distribution of dominant animals" (line 60), "For all organisms it is essential to choose suitable habitats to meet their needs for food, shelter and reproduction" (lines 47-48). Beyond presenting an a priori predictable set of ecological rules, the introduction should explain the rationale for measuring them. The revised manuscript could emphasise that the main aim was to investigate the effects of habitat fragmentation on the mechanisms of coexistence and diversity maintenance in desert rodents (lines 75-77). How was habitat fragmentation measured? Perhaps 24 sampling points were not enough for this purpose. Also, the habitat fragmentation may have changed during the 5-year study. Rodent populations and their sizes can also change over five years.

Minor comments

Page 1, lines 46 -53. A lay reader could write the same sentences without knowing about community ecology, niche theory, and habitat selection. These sentences are trivial in a scientific introduction. Do you agree? If so, consider including them in the lay abstract, and not in the main text. If you disagree, explain in the rebuttal letter what kind of cutting-edge knowledge these sentences contain.

Page 2, lines 77-80. The degree of landscape heterogeneity determines the expected differences between microhabitats. A homogeneous landscape will reduce the differences. A heterogeneous landscape will increase them. Why such a trivial hypothesis? Perhaps the aim of this work is not to test this hypothesis, but to quantify the niche overlap between four rodent species and to quantify the influence of some factors on the observed niche overlap. Quantification is a necessary task, even if the hypothesis is trivial. Another example of a trivial hypothesis is the statement "the amount of food available determines habitat choice" (lines 80-82). Quantifying such a relationship is no longer trivial, but it is a necessary task to measure the discrepancy between predicted and observed ecological niche overlap between species. The discrepancy between predictions and observations is where new advances in knowledge lie.
The introduction should make it clear that the aim is not to test or reject a set of obvious hypotheses and predictions, but to quantify the influence of a sample of factors on niche overlap. In such a context, the challenge for the introduction is to explain convincingly why it is necessary to measure these relationships. Perhaps because it is not known whether the effects are linear or non-linear, whether there are interactions (also linear, non-linear, etc.), and to what extent the interactions between apparent causes add to our knowledge of niche overlap. The casuistry of each study system may justify the need to quantify plain predictions. However, it is important that the introduction explains and convinces of this need.
Page 3, lines 86-100. Delete these sentences.

Page 3, lines 102-103. Figure 1. It is unclear whether the geographical boundaries of the Alxa desert extend beyond its southern and eastern edges. I would not leave blank areas, especially to the east and south of the study area. What was the landscape type outside the study area on these sides?
In addition, the scale of Figure 1 is much larger than necessary to highlight the study area, which is so small that, for example, it is not even possible to add the 24 sampling plots or the location of the weather station. The distribution of the 24 sampling plots is a key data in this Figure, as it shows whether the plots distribution was proportional to the abundance of the main landscape types. A straightforward solution to improve Figure 1 is to add a sub-figure showing only the study area with the 24 plots plus the meteorological station. The distance between this station and the most distant sampling point could be huge, calling into question the validity of the meteorological data for distant plots.

Page 4, lines 117-135. It is useful to draw the experimental design of a sampling plot. A sketch or picture may be sufficient to visualise what is otherwise explained in lines 117-135. Consider adding a new Figure.

Page 2, line 122. The traps were baited every day. How many days per season? How many traps were checked each day? How many plots were checked each day? Capture rates can change with, for example, rain and temperature, for example. It is difficult to understand the field work because the median distance (km) between sampling points and to the meteorological station is not shown. The distribution of distances (median or mean distance, plus either SE, SD or 95% confidence interval) between sampling plots is a key variable. A tagged individual may be recaptured at two or more sampling sites if the distance is small.

Page 4, lines 136-140. The objective of measuring soil hardness and composition was not justified in the introduction. Nor in the methods. Please add a justification for these measurements.
These lines, without establishing a logical link between plants and soil, have been placed in section 2.2.2 - Vegetation sampling. Perhaps the soil influences not only the vegetation but also the ability of rodents to build shelters. In this case, it should be in a separate section and would also make it easier to justify measuring a soil variable related to the ease of burrowing.

Page 4, lines 141-144. Why is meteorological sampling included in the vegetation section? Meteorology deserves its own section because it can quantify the interactions between other variables. Please explain why you have included meteorological variables in the analyses, as they are not measured at every sampling plot. Therefore, meteorological variability between plots is excluded, even though it may, for example, explain variability in capture rates.
Perhaps meteorological data were used to account for inter-annual (years 2017, 2018, 2019, 2020 and 2021) and seasonal (spring, summer and autumn) variability in rodent abundance, as well as annual and seasonal changes in niche overlap. The meteorological data for each year may have been averaged to obtain seasonal averages (spring, summer and autumn). Further explanation of how the weather station data were used and aggregated is needed. Please forgive this comment if the explanation is available in the main text.

Page 5, line 159. Typo: delete two of the three 'species'. The first sentence could read "…the niche overlap between species i and k;"

Page 5, lines 180-181. It is unclear how a variable can be greater than a maximum value. Please, explain.

Page 5, line 192. Table 1. Please add the units for each variable. Dimensionless variables have no units, of course.

Page 6, line 193. Populations of four rodent species were estimated using the capture-mark-recapture method in 24 sampling plots over three seasons and five years. Please include monthly and annual results in the manuscript. For example, it is not clear how many times you collected zero individuals of each species. The variability of population density in each sampling plot over the five years of the study is also not known. Populations may not have been stable. Some may have increased and others may have decreased over the 5-year study period. Population size and density should be stable over the study period. If there are changes, population sizes or densities should be reported as results to allow conclusions to be drawn about spatial niche dynamics during the study period. Please report results on rodent population size for each species and sampling plot. Seasonal changes should also be reported. The population size of at least one rodent species varied seasonally (lines 277-278). In addition, the population size of M. meriadianus was a key factor (lines 216-219). Overall, population sizes should be reported in the results. A new Figure or Table should show the annual variability of each rodent species as well as the spatial variability. Such a Figure or Table can be included in the main text or in the supplementary material.

Page 6, line 208. Figure 2. The variability of the spatial niche breathing is shown with a vertical interval. What does the line represent? The standard error, or perhaps the standard deviation. It is important to explain what the interval is because the spatial niche breath of the fourth species, Meriones meridianus, is larger than that of the other first two species, Dipus sagita and Orientallactaga sibirica. And yet the same capital letter is assigned to all three. Perhaps the letter for M. medirianus should be different. Please check the letters and explain what the vertical interval is.
The histograms show the mean of five years, or the mean of 24 sampling plots, or both. An explanation is needed of how the data from the sampling plots and the years were combined to calculate an average niche breath. The sampling plots are horizontal replicates and the years are repeated measures of each replicate. It is therefore not easy to understand how the data were combined and needs to be explained.

Page 7, line 216. Typo: delete the line.

Page 8, line 230. Figure 3. Add the units on the vertical axis. The spatial niche ranged from -5 to more than 15, but it is unclear what the units are. In Figure 2 the spatial niche ranged from 0.0 to 1.0. Why are there much larger values in Figure 3?

Page 9, line 261, Typo: Three seasons, not four.

Page 11, line 327. Delete "We believe that". The sentence can start with "The coexistence…"

Page 11, lines 327-331. The first sentences of the so-called "conclusions" are trivial. Did you expect to falsify the niche variation hypothesis in this study?  As mentioned above, the study attempts to quantify the relationships between habitat structure, population density and niche respiration, which is a commendable task. Lines 331 - 334 are good examples. However, lines 327-330 contain a truism (i.e. a statement that is obviously true) and therefore add little value to the study. The first few sentences of the conclusions can be deleted or retained, but bear in mind that results were expected to confirm the hypothesis.

Reviewer 2 Report

Comments and Suggestions for Authors

The authors live-trapped four species of desert rodents and sampled habitat characteristics (vegetation, soil, and weather) to examine spatial niche breadth of each species and key environmental factors influencing this measure, as well as degree of spatial niche overlap among species. Strengths of the study include trapping and sampling over a period of several years (2017-2021) and the comprehensive nature of the study (e.g., large number of variables). The major weaknesses include lack of coverage of two important topics: 1) a description of basic characteristics of each of the four species in the Introduction (see below); and 2) a section on study limitations in the Discussion (see below). My specific comments are detailed below by section.

Simple Summary: Given that this section is for members of the public, I would make the following changes: 

Line 17: Include location of the Alxa desert area as done in the Abstract (line 26, Inner Mongolia, China).

Lines 19-20: Include that you sampled vegetation and soil and monitored weather (because you mention at least shrub height and density in lines that follow).

Lines 22-23: Consider including common names for each of the four species.

Line 23: Note that “The” should be deleted before scientific names here (The Orientallaactaga sibirica) and throughout the manuscript.

Add a concluding statement at the very end.

Abstract: If you have room, include that you sampled vegetation and soil and monitored weather.

Line 35: What is meant by “evenness” of rodent communities?

Key words: The terms “desert” and “rodents” appear in the title, so consider replacing them with the scientific names (or common names) of the four study species.

Introduction:

Line 42: Is “set” the correct word here? Maybe “set of conditions”?

Line 46: Is “mammalian regions” correct here?

Line 74: I would start a new paragraph here that describes your study. I suggest including some basic characteristics of each of the four study species, such as adult body size, diet (e.g., omnivore, herbivore, etc.), mode of locomotion (quadrupedal or bipedal), activity pattern (e.g., all nocturnal), whether or not they hibernate in winter, timing of breeding seasons, and social and mating systems, if known. These characteristics should be included in the Introduction, which will set the stage for your coverage of them in the Discussion. 

Lines 79-80 require editing for clarity.

Materials and Methods:

Line 96, Section 2.2: Change “Data selection” to “Data collection”

Lines 103-104: Were the traps single-capture or multiple-capture?

Line 106: What categories were used to classify reproductive status?

Data on sex, reproductive status, and body size were not included in the results of this paper. Will they be presented elsewhere? If so, you might want to state this, so readers won’t expect to see coverage of these topics in the Results section.

Lines 97-100 and lines 105-106: Just to make sure I understand, live-trapping was conducted for the first 14 days of April, July, and October for 5 consecutive years (2017-2021)? 

Line 141: The name should be “Colwell.”

Results:

Line 177, 3.1.1: A title for this subsubsection is needed.

The tables and figures are nicely done.

Discussion:

Line 277: The phrase “habitats with height shrubs” requires editing.

Lines 285-286: Phrases are repeated.

A paragraph on study limitations is needed. This might include some concerns about the minimum number known alive method to calculate population size. See, for example, the following paper:

M.J.O. Pocock et al. 2004. Tapering Bias Inherent in Minimum Number Alive (MNA) Population Indices. Journal of Mammalogy, 85(5):959–962.

Conclusions:

Line 304: Consider changing “We believe” to “We suggest” 

Lines 308-311: These two sentences on spatial niche breadth report specific results and do not represent conclusions.

Note also, that in the Introduction, the impact of habitat fragmentation on the coexistence of desert rodent species is mentioned (lines 74-77) but never returned to in either the Discussion or Conclusion sections.

Comments on the Quality of English Language

Moderate editing for English grammar and word usage is needed.

Reviewer 3 Report

Comments and Suggestions for Authors

Manuscript ID: animals-2889547
Type of manuscript: Article
Title: The Spatial Niche and Influencing Factors of Desert Rodents
Authors: Xin Li, Na Zhu, Ming Ming, Lin-Lin Li, Fan Bu, Xiao-Dong Wu, Shuai
Yuan *, He-Ping Fu *

Review

Authors analyzed the desert rodents spatial niches, coexistence patterns, and influencing factors with the aim to explore the impact of habitat fragmentation, They hypothesized that the heterogeneous environment of desert region drives the occupation of different micro-habitats by various rodents, with coexistence being possible if there is a trade-off between foraging efficiency and the cost of travel.

Manuscript is interesting, and will have attraction of readers, however, it needs some revision before being accepted. Please find my comments below.

General comments

1.     Language is not sufficient, moderate edition required by native speaker. E.g., Lines 74–81, text is repeated, sentence started with “And”.

2.     Common names of species should be used with Latin names in the first occasion, E.g., Lines 91–93.

3.     When using short form, such as P.roborovskii, space must be inserted after shortened genus name: P. roborovskii. Check throughout the text

4.     Space is required before opening parenthesis for referring, e.g., Line 41 “coexistence [1–3].”, not “coexistence[1–3]. Check throughout, there are tens of such places.

5.     Use long dash for ranges, e.g., Line 145 must read “ranges between 0–1 [33].”, not “ranges between 0-1[33].”. Check throughout

6.     Check references. E.g., [17] have many authors, not only K. Mondal

Specific comments

Line 21: wording “habitat shrub height and density” is not correct. Maybe parentheses are missing?

Line 96: Data collection?

Line 114: change chapter title, now it is the same as in Line 115.

Line 120: use long dash for ranges

Line 120–125: insert space before dimension

Line 138: delete “representing the common logarithm”

Line 141: please add reference to the list

Line 142: delete one ford from “species k species”

Line 145: wrong punctuation in “levels, The”

Table 3 body: change defies to “minus” sign

Figure 4: Genus name must be separated from species name by a space

Line 330: “interest.The” space missing

Line 376: excessive space

[19] – check DOI, it is not working

[20] – mistake in DOI number

[25] – part of bibliographic information is missing

[26] – check DOI, it is not working

[28]: excessive spaces

[32] – mistypes

Line 410: bold not needed

[36] – version?

[39] – check DOI, it is not working. Spaces?

[40] – presented DOI points out to different paper

[41], [50], [51] – check DOI, these are not working and not registered in DOI.

Comments on the Quality of English Language

Language needs editing

Reviewer 4 Report

Comments and Suggestions for Authors

The authors presented research on the niche breadth and overlap of four desert rodents in Inner Mongolia. The research is interesting and useful for the conservation management of mammals living in harsh environments. Data collection, analysis and discussion of findings are nicely presented. The English language is fine. The study’s subject falls well within the scope of the journal.

Some improvements, in study design mostly, will help increase understanding by the readers.

Comments

Line 46: “mammalian regions”? I think mammalian species. Please check and amend.

Figure 1 legend: Correct to “temperate steppe”.

Lines 97-98: Give a map with the position of plots and traps.

Line 102: 7x8 refers to m? Please and unit of measuremet.

Lines 102-105: it is difficult to figure out the trap layout based of this description. A better description and the proposed map with trap design overlaid would help. Also, give the total number of traps.

Line 177: Give subsection title.

Lines 183-189 and Table 2: Calculate bootstrap confidence interval to infer significance of niche overlap.

Lines 197-208 and Figure 3: Indicate thresholds for high, very high contributions inferred by random forests in Figure 3. Which factors did you use in subsequent analyses? Those of very high contribution only? Or also those with high contribution. Be specific both in text and figure.

Figure 3: Name the y-axes.

Lines 212-213: This should have become obvious from the previous paragraph. It will if you follow the previous two comments.
